# Re-examination of the Role of Latent Variables in Sequence Modeling

**Guokun Lai**[*1]**, Zihang Dai**[*1]**, Yiming Yang**[1]**, Shinjae Yoo**[2]
[1]Carnegie Mellon University, [2]Brookhaven National Laboratory
[1]{guokun,dzihang,yiming}@cs.cmu.edu, [2]sjyoo@bnl.gov

## Abstract

With latent variables, stochastic recurrent models have achieved state-of-the-art performance in modeling sound-wave sequence. However, opposite results are also observed in other domains, where standard recurrent networks often outperform stochastic models. To better understand this discrepancy, we re-examine the roles of latent variables in stochastic recurrent models for speech density estimation. Our analysis reveals that under the restriction of fully factorized output distribution in previous evaluations, the stochastic variants were implicitly leveraging intra-step correlation but the deterministic recurrent baselines were prohibited to do so, resulting in an unfair comparison. To correct the unfairness, we remove such restriction in our re-examination, where all the models can explicitly leverage intra-step correlation with an auto-regressive structure. Over a diverse set of univariate and multivariate sequential data, including human speech, MIDI music, handwriting trajectory and frame-permuted speech, our results show that stochastic recurrent models fail to deliver the performance advantage claimed in previous work. In contrast, standard recurrent models equipped with an auto-regressive output distribution consistently perform better, dramatically advancing the state-of-the-art results on three speech datasets.

## 1 Introduction

As a fundamental problem in machine learning, probabilistic sequence modeling aims at capturing the sequential correlations in both short and long ranges. Among many possible model choices, deep auto-regressive models [1, 2] have become one of the most widely adopted solutions. Typically, a deep auto-regressive model factorizes the likelihood function of sequences in an auto-regressive manner, i.e., $p(\mathbf{x}) = \prod_{t=1}^{|\mathbf{x}|} p(x_t \mid \mathbf{x}_{<t})$. Then, a neural network (e.g. RNN) is employed to encode the conditional context $\mathbf{x}_{<t}$ into a compact hidden representation $h_t = f(\mathbf{x}_{<t})$, which is then used to define the output distribution $p(x_t \mid \mathbf{x}_{<t}) \triangleq p(x_t \mid h_t)$.

Despite the state-of-the-art (SOTA) performance in many domains [3, 4, 5, 6], the hidden representations of standard auto-regressive models are produced in a completely deterministic way. Hence, the stochastic aspects of the observed sequences can only be modeled by the output distribution, which however, usually has a simple parametric form such as a unimodal distribution or a finite mixture of unimodal distributions. A potential weakness of such simple forms is that they may not be sufficiently expressive for modeling real-world sequential data with complex stochastic dynamics.

Recently, many efforts have been made to enrich the expressive power of auto-regressive models by injecting stochastic latent variables into the computation of hidden states. Notably, relying on the variational auto-encoding (VAE) framework [7, 8], stochastic recurrent models (SRNN) have outperformed standard RNN-based auto-regressive models by a large margin in modeling raw sound-wave sequences [9, 10, 11, 12, 13, 14].

However, the success of stochastic latent variables does not necessarily generalize to other domains such as text and images. For instance, the authors [12] report that an SRNN trained by Z-Forcing lags behind a baseline RNN in language modeling. Similarly, for the density estimation of natural images, PixelCNN [15, 16, 17] consistently outperforms generative models with latent variables [18, 19, 20, 21, 22].

To better understand the discrepancy, we perform a re-examination on the role of stochastic variables in SRNN models. By carefully inspecting of the previous experiment settings for sound-wave density estimation, and systematically analyzing the properties of SRNN, we identify two potential causes of the performance gap between SRNN and RNN. Controlled experiments are designed to test each hypothesis, where we find that previous evaluations impose an unnecessary restriction of fully factorized output distributions, which has led to an unfair comparison between SRNN and RNN. Specifically, under the factorized parameterization, SRNN can still implicitly leverage the intra-step correlation, i.e., the simultaneity [23], while the RNN baselines are prohibited to do so. Meanwhile, we also observe that the posterior learned by SRNN can get outperformed by a simple hand-crafted posterior, raising serious doubt about the general effectiveness of injecting latent variables.

To provide a fair comparison, we propose an evaluation setting where both the SRNN and RNN can utilize an auto-regressive output distribution to model the intra-step correlation explicitly. Under the new setting, we re-evaluate SRNN and RNN on a diverse collection of sequential data, including human speech, MIDI music, handwriting trajectory and frame-permuted speech. Empirically, we find that sequential models with continuous latent variables fail to offer any practical benefits, despite their widely believed theoretical superiority. On the contrary, explicitly capturing the intra-step correlation with an auto-regressive output distribution consistently performs better, substantially improving the SOTA performances in modeling speech signals. Overall, these observations show that the previously reported performance "advantage" of SRNN is merely the result of a long-existing experiment bias of using factorized output distributions.

## 2   Background

In this section, we briefly review SRNN and RNN for probabilistic sequence modeling. The other approaches are summarized in appendix E. Throughout the paper, we will use bold font $\mathbf{x}$ to denote a sequence, $\mathbf{x}_{<t}$ and $\mathbf{x}_{\le t}$ to indicate the sub-sequence of first $t-1$ and $t$ elements respectively, and $x_t$ to represent the $t$-th element. Note that $x_t$ can either be a scalar or a multivariate vector. In the latter case, $x_{t,i}$ denotes the $i$-th element of the vector $x_t$.

Given a set of sequences $\mathcal{D} = \left\{ \mathbf{x}^1, \mathbf{x}^2, \cdots, \mathbf{x}^{|\mathcal{D}|} \right\}$, we are interested in building a density estimation model for sequences. A widely adapted solution is to employ an auto-regressive model powered by a neural network, and utilize MLE to perform the training:

$$\max_{\theta} \mathcal{L}_{\mathcal{D}} = \mathop{\mathbb{E}}_{\mathbf{x} \sim \mathcal{D}} \left[ \sum_{t=1}^{T_x} \log p_\theta(x_t \mid \mathbf{x}_{<t}) \right], \tag{1}$$

where $T_x$ is the length of the sequence $\mathbf{x}$. More concretely, the conditional distribution $p_\theta(x_t \mid \mathbf{x}_{<t})$ is usually jointly modeled by two sub-modules:

- The pre-defined distribution family of the output distribution $p_\theta(x_t \mid \mathbf{x}_{<t})$, such as a Gaussian, Categorical or Gaussian Mixture;
- The sequence model $f_\theta$, which encodes the contextual sequence $\mathbf{x}_{<t}$ into a compact hidden vector $h_t$;

Under this general framework, RNN and SRNN can be seen as two different instantiations of the sequence model. As we have discussed in Section 1, the computation inside RNN is fully deterministic.

To improve the model expressiveness, SRNN takes an alternative route and incorporates *continuous* latent variables into the sequence model. Typically, SRNN associates the observed data sequence $\mathbf{x}$ with a sequence of latent variables $\mathbf{z} = [z_1, \ldots, z_{T_x}]$, one for each step. With latent variables, the internal dynamics of the sequence model is not deterministic any more, offering a theoretical possibility to capture more complex stochastic patterns. However, the improved capacity comes with

a computational burden — the log-likelihood is generally intractable due to the integral:

$$\mathcal{L}_{\mathcal{D}}^{\text{SRNN}} = \mathop{\mathbb{E}}_{\mathbf{x} \sim \mathcal{D}} \left[ \log \int p_\theta(\mathbf{x}, \mathbf{z}) d\mathbf{z} \right].$$

Hence, standard MLE training cannot be performed.

To handle the intractability, SRNN utilizes the VAE framework and maximizes the evidence lower bound (ELBO) of the log-likelihood (1) for training:

$$\max_{\theta, \phi} \mathcal{F}_{\mathcal{D}} = \mathop{\mathbb{E}}_{\mathbf{x} \sim \mathcal{D}} \left[ \mathop{\mathbb{E}}_{q_\phi(\mathbf{z}|\mathbf{x})} \left( \sum_{t=1}^{T_x} \log \frac{p_\theta(x_t \mid \mathbf{z}_{\leq t}, \mathbf{x}_{<t}) p_\theta(z_t | \mathbf{z}_{<t}, \mathbf{x}_{<t})}{q_\phi(z_t \mid \mathbf{z}_{<t}, \mathbf{x})} \right) \right] \leq \mathcal{L}_{\mathcal{D}}^{\text{SRNN}}, \tag{2}$$

where $q_\phi(\mathbf{z} \mid \mathbf{x})$ is the approximate posterior distribution modeled by an encoder network. Computationally, several SRNN variants have been proposed [9, 10, 11, 12], mostly differing in how the generative distribution $p_\theta(\mathbf{x}, \mathbf{z})$ and the variational posterior $q_\phi(\mathbf{z} \mid \mathbf{x})$ are parameterized. In this work, we follow the parameterization and optimization in Z-forcing SRNN method [12], which is the one with the best performance. We include a detailed introduction for related SRNN models in appendix B.

## 3  Revisiting SRNN for Speech Modeling

### 3.1  Previous Setting for Speech Density Estimation

To compare SRNN and RNN, previous studies largely rely on the density estimation of sound-wave sequences. Usually, a sound-wave dataset consists of a collection of audio sequences with a sample rate of 16Hz, where each frame (element) of the sequence is a scalar in $[-1, 1]$, representing the normalized amplitude of the sound. Instead of treating each frame as a single step, the authors [10] propose a multi-frame setting, where every 200 consecutive frames are taken as a single step. Effectively, the data can be viewed as a sequence of 200-dimensional real-valued vectors, i.e., $x_t \in \mathbb{R}^L$ with $L = 200$. During training, every $T = 40$ steps (8,000 frames) are taken as an i.i.d. sequence to form the training set.

Under this data format, notice that the output distributions $p_\theta(x_t \mid \mathbf{x}_{<t})$ and $p_\theta(x_t \mid \mathbf{z}_{\leq t}, \mathbf{x}_{<t})$ now correspond to an $L$-dimensional random vector $x_t$. Therefore, how to parameterize this multivariate distribution can largely influence empirical performance. That said, recent approaches [11, 12] have all followed [10] to employ a fully factorized parametric form which ignores the inner dependency:

$$p_\theta(x_t \mid \mathbf{x}) \approx \prod_{i=1}^{L} p_\theta(x_{t,i} \mid \mathbf{x}_{<t}), \tag{3}$$

$$p_\theta(x_t \mid \mathbf{z}_{\leq t}, \mathbf{x}_{<t}) \approx \prod_{i=1}^{L} p_\theta(x_{t,i} \mid \mathbf{z}_{\leq t}, \mathbf{x}_{<t}). \tag{4}$$

Here, we have used the $\approx$ to emphasize this choice effectively poses an independent assumption. Despite this convenience, note that the restriction of a fully factorized form is not necessary at all. Nevertheless, we will refer to the models in Eqn. (3) and Eqn. (4), respectively, as factorized RNN (F-RNN) and factorized SRNN (F-SRNN) in the following.

To provide a baseline for further discussion, we replicate the experiments under the setting introduced above and evaluate them on three speech datasets, namely TIMIT, VCTK, and Blizzard. Following the previous work [10], we choose a Gaussian mixture to model the per-frame distribution $p_\theta(x_{t,i} \mid \mathbf{x}_{<t})$ of F-RNN, which enables a basic multi-modality.

We report the averaged test log-likelihood in Table 1. For consistency with previous results in the literature, the results of TIMIT and Blizzard are based on *sequence-level* average, while the result of VCTK is *frame-level* average. As we can see, similar to previous observations, F-SRNN outperforms F-RNN on all three datasets by a dramatic margin.

| Models | TIMIT | VCTK | Blizzard |
|--------|-------|------|----------|
| F-RNN | 32,745 | 0.786 | 7,610 |
| F-SRNN | **69,296** | **2.383** | **15,258** |

Table 1: Performance comparison on three benchmark datasets.

### 3.2 Decomposing the Advantages of Factorized SRNN

To understand why the F-SRNN outperforms F-RNN by such a large margin, it is helpful to examine the effective output distribution $p_\theta(x_t \mid \mathbf{x}_{<t})$ of F-SRNN after marginalizing out the latent variables:

$$p_\theta(x_t \mid \mathbf{x}_{<t}) = \int p_\theta(\mathbf{z}_{\leq t} \mid \mathbf{x}_{<t}) \prod_{i=1}^{L} p_\theta(x_{t,i} \mid \mathbf{z}_{\leq t}, \mathbf{x}_{<t}) d\mathbf{z}_{\leq t}. \tag{5}$$

From this particular form, we can see two potential causes of the performance gap between F-SRNN and F-RNN in the multi-frame setting:

- **Advantage under High Volatility**: By incorporating the continuous latent variable, the distribution $p_\theta(x_t \mid \mathbf{x}_{<t})$ of F-SRNN essentially forms an infinite mixture of simpler distributions (see first line of Eqn. (5)). As a result, the distribution is significantly more expressive and flexible, and it is believed to be particularly suitable for modeling high-entropy sequential dynamics [10].

  The multi-frame setting introduced above well matches this description. Concretely, since the model is required to predict the next $L$ frames all together in this setting, the long prediction horizon will naturally involve a higher uncertainty. Therefore, the high volatility of the multi-frame setting may provide a perfect scenario for SRNN to exhibit its theoretical advantage in expressiveness.

- **Utilizing the Intra-Step Correlation**: From Eqn. (5), notice that the distribution $p_\theta(x_t \mid \mathbf{x}_{<t})$ after marginalization is generally not factorized any more, due to the coupling with $\mathbf{z}$. In contrast, recall the same distribution of the F-RNN (Eqn. (3)) is fully factorized $p_\theta(x_t \mid \mathbf{x}_{<t}) = \prod_{i=1}^{L} p_\theta(x_{t,i} \mid \mathbf{x}_{<t})$. Therefore, in theory, a factorized SRNN could still model the correlation among the $L$ frames within each step, if properly trained, while the factorized RNN has no means to do so at all. Thus, SRNN may also benefit from this difference.

While both advantages could have jointly led to the performance gap in Table 1, the implications are totally different. The first advantage under high volatility is a unique property of latent-variable models that other generative models without latent variables can hardly to obtain. Therefore, if this property significantly contributes to the superior performance of F-SRNN over F-RNN, it suggests more general effectiveness of incorporating stochastic latent variables.

Quite the contrary, being able to utilize the intra-step correlation is more like an *unfair* benefit to SRNN, since it is the unnecessary restriction of fully factorized output distributions in previous experimental design that prevents RNNs from modeling the correlation. In practice, one can easily enable RNNs to do so by employing a non-factorized output distribution. In this case, it remains unclear whether this particular advantage will sustain. Motivated by the distinct implications, in the sequel, we will try to figure out how much each of the two hypotheses above actually contributes to the performance gap.

### 3.3 Advantage under High Volatility

In order to test the advantage of F-SRNN in modeling high-volatile data in isolation, the idea is to construct a sequential dataset where each step consists of a single frame (i.e., a uni-variate variable), while there exists high volatility between every two consecutive steps.

Concretely, for each sequence $\mathbf{x} \in \mathcal{D}$, we create a sub-sequence by selecting one frame from every $M$ consecutive frames, i.e., $\hat{\mathbf{x}} = [x_1, x_{M+1}, x_{2M+1}, \ldots]$ with $x_t \in \mathbb{R}$. Intuitively, a larger *stride* $M$ will lead to a longer horizon between two selected frames and hence a higher uncertainty. Moreover, since each step corresponds to a single scalar, the second advantage (i.e., the potential confounding factor) automatically disappears.

Following this idea, from the original datasets, we derive the stride-TIMIT, stride-VCTK and stride-Blizzard with different stride values $M$, and evaluate the RNN and SRNN on each of them. Again, we report the sequence- or frame-average test likelihood in Table 2.

| Model | Stride = 50 | | | Stride = 200 | | |
|---|---|---|---|---|---|---|
| | TIMIT | VCTK | Blizzard | TIMIT | VCTK | Blizzard |
| RNN | **20,655** | **0.668** | **4,607** | **4,124** | **0.177** | **-320** |
| SRNN | 14,469 | 0.605 | 3,603 | -1,137 | 0.0187 | -1,231 |

Table 2: Performance comparison on high-volatility datasets.

Surprisingly, RNN consistently achieves a better performance than SRNN in this setting. It suggests the theoretically better expressiveness of SRNN does not help that much in high-volatility scenarios. Hence, this potential advantage does not really contribute to the performance gap observed in Table 1.

### 3.4 Utilizing the Intra-Step Correlation

After ruling out the first hypothesis, it becomes more likely that being able to utilize the intra-step correlation actually leads to the superior performance of F-SRNN. However, despite the non-factorized form in Eqn. (5), it is still not clear how F-SRNN computationally captures the correlation in practice. Here, we provide a particular possibility.

Recall that in ELBO function of SRNN method (Eqn. (2)), the vector $x_t$, we hope to reconstruct at step $t$, is included in the conditional input to the posterior $q_\phi(z_t \mid \mathbf{z}_{<t}, \mathbf{x})$. With this computational structure, the encoder could theoretically leak a *subset* of the vector $x_t$ into the latent variable $z_t$, and leverage the leaked subset to predict (reconstruct) the rest elements in $x_t$. Intuitively, the procedure of using the leaked subset to predict the remained subset is essentially exploiting the dependency between the two subsets, or in other words, the correlation within $x_t$.

**Proposition 1.** *Given a vector $x_t$, we split its elements into two arbitrary disjoint subsets, the leaked subset $x_t^a$ and its complement $x_t^b = x_t \backslash x_t^a$. Assume that the latent variables and leaked subset have the same dimensionality, $|z_t| = |x_t^a|$. Define the posterior distribution as a delta function:*

$$q_\phi(z_t \mid \mathbf{z}_{<t}, \mathbf{x}) = \delta_{z_t = x_t^a} = \begin{cases} \infty, & \text{if } z_t = x_t^a \\ 0, & \text{otherwise} \end{cases}, \tag{6}$$

*We further assume $p_\theta(x_t \mid \mathbf{z}_{\leq t}, \mathbf{x}_{<t}) \approx p_\theta(x_t \mid z_t, \mathbf{x}_{<t})$. The ELBO function (Eqn. (2)) would reduce to a special case of auto-regressive factorization:*

$$\max_\theta \mathcal{L}_\mathcal{D} = \mathop{\mathbb{E}}_{\mathbf{x} \sim \mathcal{D}} \left[ \sum_{t=1}^{T_x} \Big[ \log p_\theta(x_t^a \mid \mathbf{x}_{<t}) + \log p_\theta(x_t^b \mid x_t^a, \mathbf{x}_{<t}) \Big] \right]. \tag{7}$$

This proposition can be proved by substituting the posterior distribution into the ELBO function and the detail derivation is provided in the supplementary material. Now, the second term in Eqn. (7) is conditioned on the leaked subset of $x_t^a$ to predict $x_t^b$, which is exactly utilizing the correlation between the two subsets. In other words, with a proper posterior, F-SRNN can recover a certain auto-regressive parameterization, making it possible to utilize the intra-step correlation, even with a fully factorized output distribution.

Although the analysis and construction above provide a theoretical possibility, we still lack concrete evidence to support the hypothesis that F-SRNN has significantly benefited from modeling the intra-step correlation. While it is difficult to verify this hypothesis in general, we can parameterize an RNN according to Eqn. (7), which is equivalent to an F-SRNN with a delta posterior. Therefore, by measuring the performance of this special RNN, we can get a *conservative* estimate of how much modeling the intra-step correlation can contribute to the performance of F-SRNN.

To finish the special RNN idea, we still need to specify how $x_t$ is split into $x_t^a$ and $x_t^b$. Here, we consider two methods with different intuitions:

- **Interleaving**: The first method takes one out of every $U$ elements to construct $x_t^a = \{x_{t,1}, x_{t,U+1}, x_{t,2U+1}, \dots\}$. Essentially, this method interleaves the two subsets $x_t^a$ and $x_t^b$. In the extreme case of $U = 2$, $x_t^a$ includes the odd elements of $x_t$ and $x_t^b$ the even ones. Hence, when predicting an even element $x_{t,2k} \in x_t^b$, the output distribution is conditioned on both the elements to the left $x_{t,2k-1}$ and to the right $x_{t,2k+1}$, making the problem much easier.
- **Random**: The second method simply uniformly selects $V$ random elements from $x_t$ to form $x_t^a$, and leaves the rest for $x_t^b$. Intuitively, this can be viewed as an informal "lower bound" of performance gain through modeling the intra-step correlation.

| Models | TIMIT | VCTK | Blizzard |
|---|---|---|---|
| F-RNN | 32,745 | 0.786 | 7,610 |
| F-SRNN | 69,296 | **2.383** | 15,258 |
| $\delta$-RNN ($U = 2$) | 70,900 | 2.027 | **15,306** |
| $\delta$-RNN ($U = 3$) | **72,067** | 2.262 | 15,284 |
| $\delta$-RNN ($V = 50$) | 66,122 | 2.199 | 14,389 |
| $\delta$-RNN ($V = 75$) | 66,453 | 2.120 | 14,585 |

Table 3: Performance comparison between $\delta$-RNN and F-SRNN. Note that a smaller $U$ corresponds to leaking more elements.

Since the parametric form Eqn. (7) is derived from a delta posterior, we will refer to the special RNN model as $\delta$-RNN. Based on the two split methods, we train $\delta$-RNN on TIMIT, VCTK and Blizzard with different values of $U$ and $V$. The results are summarized in Table 3. As we can see, when the interleaving split scheme is used, $\delta$-RNN significantly improves upon F-RNN and becomes very competitive with F-SRNN. Specifically, on TIMIT and Blizzard, $\delta$-RNN can even outperform F-SRNN in certain cases. More surprisingly, the $\delta$-RNN with the random-copy scheme can also achieve a performance that is very close to that of F-SRNN, especially compared to F-RNN.

Recall that $\delta$-RNN is equivalent to employing a manually designed delta posterior that can only copy but never compresses (auto-encodes) the information in $x_t$. As a result, compared to a posterior that can learn to compress information, the delta posterior will involve a higher KL cost when leaking information through the posterior. Furthermore, the correlation between historical latent variables and outputs is ignored. It would decrease the model capacity of $\delta$-RNN. Despite these disadvantages, $\delta$-RNN is still able to match or even surpasses the performance of F-SRNN, suggesting the learned posterior in F-SRNN is far from satisfying. Quite contrary to that, the limited performance gap between F-SRNN and the random copy baseline raises a serious concern about the effectiveness of current variational inference techniques.

Nevertheless, putting the analysis and empirical evidence together, we can conclude that the performance advantage of F-SRNN in the multi-frame setting can be entirely attributed to the second cause. That is, under the factorized constraint in previous experiments, F-SRNN can still implicitly leverage the intra-step correlation, while F-RNN is prohibited to do so. However, as we have discussed earlier in Section 3.2, this is essentially an unfair comparison. More importantly, the claimed superiority of SRNN over RNN may be misleading, as it is unclear whether performance advantage of SRNN will sustain or not when a non-factorized output distribution is employed to capture the intra-step correlation explicitly.

As far as we know, no previous work has carefully compared the performance of SRNN and RNN when non-factorized output distribution is allowed. On the other hand, as shown in Table 3, by modeling the multivariate simultaneity in the simplest way, $\delta$-RNN can achieve dramatic performance improvement. Motivated by the huge potential as well as the lack of a systematic study, we will next include non-factorized output distributions in our consideration, and properly re-evaluate SRNN and RNN for multivariate sequence modeling.

## 4 Proper Multivariate Sequence Modeling with or without Latent Variables

### 4.1 Avoiding the Implicit Data Bias

In this section, we aim to eliminate any experimental bias and provide a proper evaluation of SRNN and RNN for multivariate sequence modeling. Apart from the "model bias" of employing fully factorized output distributions we have discussed, another possible source of bias is actually the experimental data. For example, as we discussed in Section 3.1, the multi-frame speech sequences are constructed by reshaping $L$ consecutive real-valued frames into $L$-dimensional vectors. Consequently, elements within each step $x_t$ are simply temporally correlated with a natural order, which would favor a model that recurrently process each element from $x_{t,1}$ to $x_{t,L}$ with parameter sharing.

Thus, to avoid such "data bias", besides speech sequences, we additionally consider three more types of multivariate sequences with different patterns of intra-step correlation, they are MIDI sound sequence data (including Muse and Nottingham datasets), handwriting trajectory data (IAM-OnDB)

| Models | TIMIT | VCTK | Blizzard | Muse | Nottingham | IAM-OnDB | Perm-TIMIT |
|---|---|---|---|---|---|---|---|
| VRNN$^\dagger$ | 28,982 | - | 9,392 | - | - | 1384 | - |
| SRNN$^\dagger$ | 60,550 | - | 11,991 | -6.28 | -2.94 | - | - |
| Z-Forcing$^\dagger$ | 70,469 | - | 15,430 | - | - | - | - |
| SWaveNet$^{\dagger\ddagger}$ | 72,463 | - | 15,708 | - | - | 1301 | - |
| STCN$^{\dagger\ddagger}$ | 77,438 | - | 17,670 | - | - | **1796** | - |
| F-RNN | 32,745 | 0.786 | 7,610 | -6.991 | -3.400 | 1397 | 25,679 |
| F-SRNN | 69,296 | 2.383 | 15,258 | -6.438 | -2.811 | 1402 | 67,613 |
| $\delta$-RNN-random | 66,453 | 2.199 | 14,585 | -6.252 | -2.834 | N/A | 61,103 |
| RNN-flat | **117,721$^\star$** | **3.2173$^\star$** | **22,714$^\star$** | -5.251 | -2.180 | N/A | 15,763 |
| SRNN-flat | 109,284 | 3.2062 | 22,290 | -5.616 | -2.324 | N/A | 14,278 |
| RNN-hier | 109,641 | 3.1822 | 21,950 | **-5.161** | **-2.028** | 1440 | **95,161** |
| SRNN-hier | 107,912 | 3.1423 | 21,845 | -5.483 | -2.065 | 1395 | 94,402 |

Table 4: Performance comparison on a diverse set of datasets. The models with $^\dagger$ indicate that the performances are directly copied from previous publications. Numbers with $^\star$ indicate the state-of-the-art performances. N/A suggests the model is not application on the dataset. The models with $^\ddagger$ have other architectures than recurrent neural network as the backbone.

and the Perm-TIMIT dataset. The Perm-TIMIT is a variant of multivariate TIMIT dataset. It permutes the elements within each time step, which is designed to remove the temporal bias. We include the detail information of these datasets in Appendix C.

## 4.2 Modeling Simultaneity with Auto-Regressive Decomposition

With proper datasets, we now consider how to construct a family of non-factorized distributions that (1) can be easily integrated into RNN and SRNN as the output distribution, and (2) are reasonably expressive for modeling multivariate correlations. Among many possible choices, the most straight-forward choice would be the auto-regressive parameterization. Compared to other options such as the normalizing flow or Markov Random Field (e.g. RBM), the auto-regressive structure is conceptually simpler and can be applied to both discrete and continuous data with full tractability. In light of these benefits, we choose to follow this simple idea, and decompose the output distribution of the RNN and SRNN, respectively, as

$$p_\theta(x_t \mid \mathbf{x}_{<t}) = \prod_{i=1}^{L} p_\theta(x_{t,i} \mid \mathbf{x}_{<t}, x_{t,<i}), \tag{8}$$

$$p_\theta(x_t \mid \mathbf{z}_{\leq t}, \mathbf{x}_{<t}) = \prod_{i=1}^{L} p_\theta(x_{t,i} \mid \mathbf{z}_{\leq t}, \mathbf{x}_{<t}, x_{t,<i}). \tag{9}$$

Notice that although we use the natural decomposition order from smallest index to largest one, this particular order is generally *not* optimal for modeling multivariate distributions. A better choice could be adapting the orderless training previously explored in literature [2]. But for simplicity, we will stick to this simple approach.

Given the auto-regressive decomposition, a natural neural instantiation would be a recurrent *hierarchical model* that utilizes a two-level architecture to process the sequence:

- Firstly, a high-level RNN or SRNN is employed to encode the multivariate steps $\mathbf{x} = [x_1, \ldots, x_T]$ into a sequence of high-level hidden vectors $\mathbf{h} = [h_1, \ldots, h_T]$, which follows exactly the same as the computational procedure used in F-RNN and F-SRNN . Recall that, in the case of SRNN, the computation of high-level vectors involves sampling the latent variables.
- Based on the high-level representations, for each multivariate step $x_t$, another neural model $f_{\text{low}}$ will take both the elements $[x_{t,1}, \cdots, x_{t,L}]$ and the high-level vector $h_t$ as input, and auto-regressively produce a sequence of low-level hidden vectors $[g_{t,1}, \cdots, g_{t,L}]$ where $g_{t,i} = f_{\text{low}}(x_{t,<i}, h_t)$. They can be then used to form the per-element output distributions in Eqn. (8) and (9).

In practice, the low-level model could simply be an RNN or a causally masked MLP [24], depending on our prior about the data. For convenience, we will refer to the hierarchical models as RNN-hier and SRNN-hier.

In some cases where all the elements within a step share the same statistical type, such as on the speech or MIDI dataset, one may alternatively consider a *flat model*. As the name suggests, the flat model will break the boundary between steps and flatten the data into a new uni-variate sequence, where each step is simply a single element. Then, the new uni-variate sequence can be directly fed into a standard RNN or SRNN model, producing each conditional factor in Eqn. (8) and (9) in an auto-regressive manner. Similarly, this class of RNN and SRNN will be referred to as RNN-flat and SRNN-flat, respectively. Compared to the hierarchical model, the flat variant implicitly assumes a sequential continuity between $x_{t,L}$ and $x_{t+1,1}$. Since this inductive bias matches the characteristics of multi-frame speech sequences, we expect the flat model to perform better in this case.

### 4.3 Experiment Results

Based on the seven datasets, we compare the performance of the models introduced above. To provide a random baseline, we include the $\delta$-RNN with the random split scheme in the comparison. Moreover, previous results, if exist, are also presented to provide additional information. For a fair comparison, we make sure all models share the same parameter size. For more implementation details, please refer to the Supplementary D as well as the source code[1]. We also include the running time comparison in Appendix 4.4. Finally, the results are summarized in Table 4, where we make several important observations.

Firstly, on the speech and MIDI datasets, models with auto-regressive (lower-half) output distributions obtain a dramatic advantage over models with fully factorized output distributions (upper-half), achieving new SOTA results on three speech datasets. This observation reminds us that, besides capturing the long-term temporal structure across steps, how to properly model the intra-step dependency is equally, if not more, crucial to the practical performance.

Secondly, when the auto-regressive output distribution is employed (lower-half), the non-stochastic recurrent models consistently outperform their stochastic counterparts across all datasets. In other words, the advantage of SRNN completely disappears once a powerful output distribution is used. Combined with the previous observation, it verifies our earlier concern that the so-called superiority of F-SRNN over F-RNN is merely a result of the biased experiment design in previous work.

In addition, as we expected, when the inductive bias of the flat model matches the characteristics of speech data, it will achieve a better performance than the hierarchical model. Inversely, when the prior does not match data property on the other datasets, the hierarchical model is always better. In the extreme case of permuted TIMIT, the flat model even falls behind factorized models, while the hierarchical model achieves a very decent performance that is even much better than what F-SRNN can achieve on the original TIMIT. This shows that the hierarchical model is usually more robust, especially when we don't have a good prior.

Overall, we don't find any advantage of employing stochastic latent variables for multivariate sequence modeling. Instead, relying on a full auto-regressive solution yields better or even state-of-the-art performances. Combined with the observation that $\delta$-RNN-random can often achieve a competitive performance to F-SRNN, we believe that the theoretical advantage of latent-variable models in sequence modeling is still far from fulfilled, if ultimately possible. In addition, we suggest future development along this line compare with the simple but extremely robust baselines with an auto-regressive output distribution.

### 4.4 Training Time Comparison

Here, we report the training time of different methods in TIMIT dataset. The running times of training models for 40k updating steps on TIMIT are summarized in Table 5. The input length indicates the sample length of input during the training phrase. Admittedly, modeling the intra-step correlation (\*-hier and \*-flat model) would require extra computation time. Hence, this leads to a trade-off between quality and speed. Ideally, latent-variable models would provide a solution close to the sweet point of this trade-off. However, in our experiment, we find a simple hierarchical auto-regressive model trained with a shorter input length could already achieve significantly better performance with a comparable computation time (RNN-hier vs. F-SRNN in Table 5).

| Input Length | | | | 8000 | | | | 1000 |
|---|---|---|---|---|---|---|---|---|
| Model Name | F-RNN | F-SRNN | $\delta$-RNN | RNN-hier | SRNN-hier | RNN-flat | SRNN-flat | RNN-hier |
| Training Time | 0.54h | 0.94h | 0.90h | 9.92h | 12.52h | 37.48h | 42.26h | 1.7h |
| Log-Likelihood | 32,745 | 69,296 | 66,453 | 109,641 | 107,912 | 117,721 | 109,284 | 101,713 |

Table 5: Training time comparison between various models.

## 5 Conclusion and Discussion

In summary, our re-examination reveals a misleading impression on the benefits of latent variables in sequence modeling. From our empirical observation, the main effect of latent variables is only to provide a mechanism to leverage the intra-step correlation, which is however, not as powerful as employing the straightforward auto-regressive decomposition. It remains unclear what leads to the significant gap between the theoretical potential of latent variables and their practical effectiveness, which we believe deserves more research attention. Meanwhile, given the large gain of modeling simultaneity, using sequential structures to better capture local patterns is another good future direction in sequence modeling.

## Acknowledgment

This work is supported in part by the National Science Foundation (NSF) under grant IIS-1546329 and by DOE-Office of Science under grant ASCR #KJ040201.

## Footnotes

[1]https://github.com/zihangdai/reexamine-srnn

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
