[Supplementary Material · appendix.pdf]

## A  Proof of Proposition 1

*Proof.* To facilitate the proof, we first rewrite the ELBO in Eqn. (2) in terms of the reconstruction and the KL term:

$$\mathcal{F}_{\mathcal{D}} = \mathbb{E}_{\mathbf{x} \sim \mathcal{D}} \left[ \sum_{t=1}^{T} \big( \text{Recon}_{\mathbf{x},t} + \text{KL}_{\mathbf{x},t} \big) \right], \quad \text{where}$$

$$\text{Recon}_{\mathbf{x},t} = \mathbb{E}_{q_\phi(z_t|\mathbf{x})} \big[ \log p_\theta(x_t \mid \mathbf{z}_{\leq t}, \mathbf{x}_{<t}) \big], \tag{10}$$

$$\text{KL}_{\mathbf{x},t} = -\text{KL}\big( q_\phi(z_t \mid \mathbf{x}) \big\| p_\theta(z_t \mid \mathbf{x}_{<t}) \big). \tag{11}$$

Then we substitute the delta posterior (Eqn. (6)) into both Recon and KL terms. Eqn. (10) and (11) can be simplified into

$$\text{Recon}_{\mathbf{x},t}^{\delta} = \log p_\theta(x_t^a \mid x_t^a, \mathbf{x}_{<t}) + \log p_\theta(x_t^b \mid x_t^a, \mathbf{x}_{<t}),$$

$$\text{KL}_{\mathbf{x},t}^{\delta} = -\log q_\phi(x_t^a \mid \mathbf{x}) + \log p_\theta(x_t^a \mid \mathbf{x}_{<t}).$$

Here, we also use assumption $p_\theta(x_t \mid \mathbf{z}_{\leq t}, \mathbf{x}_{<t}) \approx p_\theta(x_t \mid z_t, \mathbf{x}_{<t})$ to simplify formula. Here the terms $\log p_\theta(x_t^a \mid x_t^a, \mathbf{x}_{<t})$ and $-\log q_\phi(x_t^a \mid \mathbf{x})$ can be canceled out. Then we can rewrite the ELBO function as,

$$\max_{\theta} \mathcal{L}_{\mathcal{D}} = \mathbb{E}_{\mathbf{x} \sim \mathcal{D}} \left[ \sum_{t=1}^{T_x} \Big[ \log p_\theta(x_t^a \mid \mathbf{x}_{<t}) + \log p_\theta(x_t^b \mid x_t^a, \mathbf{x}_{<t}) \Big] \right]. \tag{12}$$

From another perspective, the form of Eqn. (12) is equivalent to a particular auto-regressive factorization of the likelihood function,

$$p_\theta^{\delta}(x_t \mid \mathbf{x}_{<t}) = p_\theta(x_t^a \mid \mathbf{x}_{<t}) p_\theta(x_t^b \mid x_t^a, \mathbf{x}_{<t})$$

$$\approx \prod_{x_{t,i} \in x_t^a} p_\theta(x_{t,i} \mid \mathbf{x}_{<t}) \prod_{x_{t,i} \in x_t^b} p_\theta(x_{t,i} \mid x_t^a, \mathbf{x}_{<t}).$$

□

## B  Different Variants of Stochastic Recurrent Neural Networks

This section detail our parameterization and recent publications [9, 10, 11, 12] of the stochastic recurrent network models.

For stochastic recurrent neural networks, the generic decomposition of generative distribution shared by previous methods has the form:

$$p(\mathbf{x}, \mathbf{z}) = \prod_{t=1}^{T} p(x_t \mid \mathbf{z}_{\leq t}, \mathbf{x}_{<t}) p(z_t \mid \mathbf{z}_{<t}, \mathbf{x}_{<t}),$$

where each new step $(x_t, z_t)$ depends on the entire history of the observation $\mathbf{x}_{<t}$ and the latent variables $\mathbf{z}_{<t}$. Similarly, for the approximate posterior distribution, all previous approaches can be unified under the form

$$q(\mathbf{z} \mid \mathbf{x}) = \prod_{t=1}^{T} p(z_t \mid \mathbf{z}_{<t}, \mathbf{x}).$$

Given the generic forms, various parameterizations with different independence assumptions have been introduced:

• STORN [9]: This parameterization makes two simplifications. Firstly, the prior distribution $p(z_t \mid \mathbf{z}_{<t}, \mathbf{x}_{<t})$ is assumed to be context independent, i.e.,

$$p(z_t \mid \mathbf{z}_{<t}, \mathbf{x}_{<t}) \approx p(z_t).$$

Secondly, the posterior distribution is simplified as

$$q(z_t \mid \mathbf{x}, \mathbf{z}_{<t}) \approx q(z_t \mid \mathbf{x}_{<t}),$$

436   which drops both the dependence on the future information $\mathbf{x}_{\geq t}$ as well as that on sub-sequence of
437   previous latent variables $\mathbf{z}_{<t}$.

438   Despite the simplification in the prior, STORN imposes no independence assumption on the output
439   distribution $p(x_t \mid \mathbf{z}_{\leq t}, \mathbf{x}_{<t})$. Specifically, an RNN is used to capture the two conditional factors
440   $\mathbf{z}_{\leq t}, \mathbf{x}_{<t}$:

$$p(x_t \mid \mathbf{z}_{\leq t}, \mathbf{x}_{<t}) \triangleq p(x_t \mid h_t),$$
$$h_t = \text{RNN}([x_{t-1}, z_t], h_{t-1}).$$

441   Notice that, the RNN is capable of modeling the correlation among the latent variables $\mathbf{z}_{\leq t}$ and
442   encodes the information into $h_t$.

443   • VRNN [10]: This parameterization eliminates some independence assumptions in STORN. Firstly,
444   the prior distribution becomes fully context dependent via a context RNN:

$$p(z_t \mid \mathbf{z}_{<t}, \mathbf{x}_{<t}) \triangleq p(z_t \mid v_{t-1}), \quad \text{where}$$
$$v_{t-1} = \text{RNN}([x_{t-1}, z_{t-1}], v_{t-2}).$$

445   Notice that $v_{t-1}$ is dependent on all previous latent variables $\mathbf{z}_{<t}$. Hence, there are no independence
446   assumptions involved in the prior distribution. However, notice that the computation of $\mathbf{v} =$
447   $[v_1, \cdots, v_T]$ cannot be parallelized due to the dependence on the latent variable as an input.

448   Secondly, compared to STORN, the posterior in VRNN additionally depends on the previous latent
449   variables $\mathbf{z}_{<t}$:

$$q(z_t \mid \mathbf{z}_{<t}, \mathbf{x}) \approx q(z_t \mid \mathbf{z}_{<t}, \mathbf{x}_{<t}) \triangleq q(z_t \mid v_{t-1}),$$

450   where $v_{t-1}$ is the same forward vector used to construct the prior distribution above. However, the
451   posterior still does not depend on the future observations $\mathbf{x}_{\geq t}$.

452   Finally, the output distribution is simply constructed as

$$p(x_t \mid \mathbf{z}_{\leq t}, \mathbf{x}_{<t}) \triangleq p(x_t \mid h_t), \quad \text{where}$$
$$h_t = [v_{t-1}, z_t].$$

453   • SRNN [11]: The abbreviation of this model is the same as the one used for the stochastic recurrent
454   neural network in the main body of our paper, but they do not stand for one thing. Compared to
455   VRNN, SRNN (1) introduces a Markov assumption into the latent-to-latent dependence and (2)
456   makes the posterior condition on the future observations $\mathbf{x}_{\geq t}$.

457   Specifically, SRNN employs two RNNs, one forward and the other backward, to consume the
458   observation sequence from the two different directions:

$$\overrightarrow{v_t} = \overrightarrow{\text{RNN}}(x_t, \overrightarrow{v_{t-1}}),$$
$$\overleftarrow{v_t} = \overleftarrow{\text{RNN}}([x_t, \overrightarrow{v_{t-1}}], \overleftarrow{v_{t+1}}).$$

459   From the parametric form, notice that $\overleftarrow{v_t}$ is always conditioned on the entire observation $\mathbf{x}$, while
460   $\overrightarrow{v_t}$ only has access to $\mathbf{x}_{\leq t}$.

461   Then, the prior and posterior are respectively formed by

$$p(z_t \mid \mathbf{z}_{<t}, \mathbf{x}_{<t}) \approx p(z_t \mid z_{t-1}, \mathbf{x}_{<t}) \triangleq p(z_t \mid \overrightarrow{v_{t-1}}, z_{t-1}),$$
$$q(z_t \mid \mathbf{z}_{<t}, \mathbf{x}) \approx q(z_t \mid z_{t-1}, \mathbf{x}) \triangleq q(z_t \mid \overleftarrow{v_t}, z_{t-1}),$$

462   where the $\approx$ indicates the aforementioned Markov assumption. In other words, given the sampled
463   value of $z_{t-1}$, $z_t$ is independent of $\mathbf{z}_{<t-1}$.

464   Finally, the output distribution of SRNN also involves the same simplification:

$$p(x_t \mid \mathbf{z}_{\leq t}, \mathbf{x}_{<t}) \approx p(x_t \mid z_t, \mathbf{x}_{<t}) = p(x_t \mid h_t), \text{where}$$
$$h_t = [\overrightarrow{v_{t-1}}, z_t].$$

465   • Z-Forcing SRNN [12]: By feeding the latent variable as an additional input into the forward
466   RNN, an approach similar to the VRNN, this parameterization successfully removes the Markov
467   assumption in SRNN.

Specifically, the computation goes as follows:

$$\overleftarrow{v_t} = \overleftarrow{\text{RNN}}(x_t, \overleftarrow{v_{t+1}}),$$
$$\overrightarrow{v_t} = \overrightarrow{\text{RNN}}([x_t, z_t], \overrightarrow{v_{t-1}}),$$

where the $z_t$ is sampled from either the prior or posterior:

$$p(z_t \mid \mathbf{z}_{<t}, \mathbf{x}_{<t}) \triangleq p(z_t \mid \overrightarrow{v_{t-1}}),$$
$$q(z_t \mid \mathbf{z}_{<t}, \mathbf{x}) \triangleq q(z_t \mid [\overrightarrow{v_{t-1}}, \overleftarrow{v_t}]).$$

Notice that, since $\overrightarrow{v_{t-1}}$ relies on $\mathbf{z}_{<t}$ in a deterministic manner, there is no Markov assumption anymore when $\overrightarrow{v_{t-1}}$ is used to construct the prior and posterior.

The same property also extends to the output distribution, which has the same parametric form as SRNN although the $\overrightarrow{v_{t-1}}$ contains different information:

$$p(x_t \mid \mathbf{z}_{\leq t}, \mathbf{x}_{<t}) = p(x_t \mid h_t), \text{where}$$
$$h_t = [\overrightarrow{v_{t-1}}, z_t].$$

In the main body of this paper, the Z-forcing SRNN is used as the parameterization of SRNN. We also follow its optimization process. Moreover, in our experiments, we find that, by dropping the dependency between historical latent variables and output, the model can be trained faster and achieve better performance in some cases. More specifically, the prior and posterior is computed in the following manner,

$$p(z_t \mid \mathbf{z}_{<t}, \mathbf{x}_{<t}) \approx p(z_t \mid \mathbf{x}_{<t}) \triangleq p(z_t \mid \overrightarrow{v_{t-1}}),$$
$$q(z_t \mid \mathbf{x}, \mathbf{z}_{<t}) \approx q(z_t \mid \mathbf{x}) \triangleq q(z_t \mid [\overrightarrow{v_{t-1}}, \overleftarrow{v_t}]),$$

where the forward and backward vectors are both computed separately in a single pass:

$$\overrightarrow{v_t} = \overrightarrow{\text{RNN}}(x_t, \overrightarrow{v_{t-1}}),$$
$$\overleftarrow{v_t} = \overleftarrow{\text{RNN}}(x_t, \overleftarrow{v_{t+1}}),$$

Then the emission distribution is computed as:

$$p(x_t \mid \mathbf{z}_{\leq t}, \mathbf{x}_{<t}) = p(x_t \mid h_t) \quad \text{where,}$$
$$h_t = \text{RNN}([\overrightarrow{v_{t-1}}, z_t], h_{t-1}),$$

For all experiment cases using SRNN in this paper, we run both Z-forcing SRNN and the above simplified version, and report the best performance. The hyper-parameter choice and implementation details are included in Appendix D.

## C  Detail Information about Datasets in Section 5

At first, we provide the details about the three additional types of data used in the experiment of Section 5.

- The first type is the MIDI sound sequence introduced in [23]. Each step of the MIDI sound sequence is 88-dimensional binary vector, representing the activated piano notes ranging from A0 to C8. Intuitively, to make the MIDI sound musically plausible, there must be some correlations among the notes within each step. However, different from the multi-frame speech data, the correlation structure is not temporal any more.

  To avoid the unnecessary complication due to overfitting, we utilize the two relatively larger datasets, namely the Muse (orchestral music) and Nottingham (folk tunes). Following earlier work [23], we report step-averaged log-likelihood for these two MIDI datasets.

- The second one we consider is the widely used handwriting trajectory dataset, IAM-OnDB. Each step of the trajectory is represented by a 3-dimension vector, where the first dimension is of a binary value, indicating whether the pen is touching the paper or not, and the second and third dimensions are the coordinates of the pen given it is on the paper. Different from other datasets, the dimensionality of each step in IAM-OnDB is significantly lower. Hence, it is reasonable to believe the intra-step structure is relatively simpler here. Following earlier work [10], we report sequence-averaged log-likelihood for the IAM-OnDB dataset.

- The last type is actually a synthetic dataset we derive from TIMIT. Specifically, we maintain the multi-frame structure of the speech sequence but permute the frames in each step with a predetermined random order. Intuitively, this can be viewed as an extreme test of a model's capability of discovering the underlying correlation between frames. Ideally, an optimal model should be able to discover the correct sequential order and recover the same performance as the original TIMIT. For convenience, we will call this dataset Perm-TIMIT.

Below is the dataset statistic information.

| Datasets | Number of Steps | Frames / Step |
|----------|-----------------|---------------|
| TIMIT | 1.54M | 200 |
| VCTK | 12.6M | 200 |
| Blizzard | 90.5M | 200 |
| Perm-TIMIT | 1.54M | 200 |
| Muse | 36.1M | 88 |
| Nottingham | 23.5M | 88 |
| IAM-OnDB | 7.63M | 3 |

Table 5: Statistics of the datasets in consideration.

The dataset statistic is summarized in Table 5. "Frame / Step" indicates the dimension of the vector $x_t$ at each time stamp. "Number of Steps" is the total length for the multivariate sequence.

## D  Experiment Details

| Domains | Speech | MIDI | Handwriting |
|---------|--------|------|-------------|
| F-RNN | 17.41M | 0.57M | 0.93M |
| F-SRNN | 17.53M | 2.28M | 1.17M |
| $\delta$-RNN-random | 18.57M | 0.71M | N/A |
| RNN-flat | 16.86M | 1.58M | N/A |
| SRNN-flat | 16.93M | 2.24M | N/A |
| RNN-hier | 17.28M | 1.87M | 0.97M |
| SRNN-hier | 17.25M | 3.05M | 1.02M |

Table 6: The parameter numbers of all implemented methods.

In the following, we will provide more details about our implementation. Firstly, Table 6 reports the parameter size of all models compared in Table 4. For data domains with enough data (i.e., speech and handwriting), we ensure the parameter size is about the same. On the smaller MIDI dataset, we only make sure the RNN variants do not use more parameters than SRNNs do.

For all methods, we use the Adam algorithm [25] as the optimizer with learning rate 0.001. The cosine schedule [26] is used to anneal the learning rate from 0.001 to 0.000001 during the training process. The batch size is set to 32 for TIMIT, 128 for VCTK and Blizzard, 16 for Muse, Nottingham, and 32 for IAM-OnDB. The total number of training steps is 20k for Muse, Nottingham, and IAM-OnDB, 40k for TIMIT, 80k for VCTK, 160K for Blizzard. For all SRNN variants, we follow previous work to employ the KL annealing strategy, where the coefficient on the KL term is increased from 0.2 to 1.0 by an increment of 0.00005 after each parameter update [12]. Because our SRNN parameterization uses Z-forcing framework, the $\alpha$ and $\beta$ value for its auxiliary loss is searched from the set $\{0, 0.0025, 0.005\}$.

For RNN-hier and SRNN-hier models, we use the RNN as the implementation of the low-level auto-regressive factorization function in the speech datasets, including TIMIT, VCTK, and Blizzard. For other datasets, we use the masked MLP as the low-level auto-regressive factorization function.

For the architectural details such as the number of layers and hidden dimensions used in this study, we refer the readers to the accompanied source code.

## E    Training Time Comparison

Here, we report the training time of different methods in TIMIT dataset. The results is illustrated in Table 7. $T_x$ is stands for the number of time stamps. $L$ is the dimension of each time stamp. All experiments are run on NVIDIA GTX 1080Ti.

| Model | $T_x$ | $L$ | **Time** |
|---|---|---|---|
| F-RNN | 40 | 200 | 0.86h |
| F-SRNN | 40 | 200 | 1.36h |
| RNN-flat | 1,000 | 1 | 20.02h |
| SRNN-flat | 1,000 | 1 | 62.86h |
| RNN-hier | 40 | 200 | 1.55h |
| SRNN-hier | 40 | 200 | 2.40h |

Table 7: Training time comparison between the models in Table 4

## F    Related Work

In the field of probabilistic sequence modeling, many efforts prior to deep learning have been devoted to State Space Models [27], such as the Hidden Markov Model [28] with discrete states and the Kalman Filter [29] whose states are continuous.

Recently, the focus has shifted to deep sequential models, including tractable deep auto-regressive models without any latent variable and deep stochastic models that combine the powerful nonlinear computation of neural networks and the stochastic flexibility of latent-variable models. The recurrent temporal RBM [30] and RNN-RBM [23] are early examples of how latent variables can be incorporated into deep neural networks. After VAE is introduced, the stochastic back-propagation makes it easy to combine the deep neural networks and latent-variable models, leading to stochastic recurrent models introduced in Section 1, temporal sigmoid belief networks [31], deep Kalman Filters [32], deep Markov Models [33], Kalman variational auto-encoders [34] and many other variants. The authors [35] provide a general discussion on how the classic graphical models and deep neural networks can be combined.