[Reviews · NeurIPS 2019]

Reviewer 1



This paper is well written and easy to follow. The experimental settings looks convincing to me, and the analysis and results are interesting. To me, this is a good piece of work on understanding stochastic models for sequence modeling. I do not fully understand section 3.4. From Eq(2), the auxiliary posterior is q_\phi(z_t | z_{

Reviewer 2



The authors discuss the role of latent variable models in sequence models where multiple observations of the time series are modeled at once using a factorized form which assumes conditional independence. This assumption is almost surely violated in practice, thus limiting the performance of such models. When the sequence model is provided with latent variables it is possible to account for the correlation structure of the likely correlated observations within a time window, thus resulting in better performance compared to models without latent variables. Results on multiple datasets demonstrate this intuition. Though the analysis presented by the authors is clear, well motivated and justified, the paper seems to downplay the importance and motivation of sequence models that consider multiple observations at once in a windowed manner, and how sequence models with stochastic (latent) variables by their ability to capture correlation structure alleviate some of the issues associated with windowing, i.e., the conditional independence assumption. The above being considered, the results in Table 4 are not surprising, however, for full context they should be presented alongside with runtimes, relegated to the supplementary material. Post rebuttal: The authors have addressed my concerns about the context of the results, runtimes and the trade-off between computational cost and performance.

Reviewer 3



- The authors verified the effectiveness of latent variables in SRNN for speech density estimation. They point out that the performance advantage of F-SRNN could be entirely attributed to implicitly utilizing the intra-step correlation. Their experimental results show that under the non-factorized output setting, no benefit of latent variables can be observed, the straightforward auto-regressive approach demonstrates superior performance. The authors performed a thorough analysis and discussion on the problem setting and give reasonable assumptions. They carefully conducted the empirical experiments and the results are convincing. - The paper offers some meaningful recommendations such as: besides capturing the long-term temporal structure across steps, properly modeling the intra-step dependency is also crucial to the practical performance. - I was surprised at the massive increase in the auto-regressive results in column 2 of Table 4. Therefore, it is somewhat uncertain to determine the difficulty and importance of these improvements. - Overall, this is a well-written paper. The language is easy to understand, and its results are highly reproducible. Therefore I recommend accepting for publication following minor revision.

[Author Response · NeurIPS 2019]

Firstly, we thank all reviewers for the helpful comments and suggestions.

**To Reviewer 2:**

$q(z_t|z_{t-1}, \mathbf{x}_{<t})$ in Eq (2) is a typo. The correct one should be $q(z_t|z_{t-1}, \mathbf{x})$, which is derived from the autoregressive
factorization of $q(\mathbf{z}|\mathbf{x})$, $q(\mathbf{z}|\mathbf{x}) = \prod_{t=1}^{T} q(z_t|z_{t-1}, \mathbf{x})$. Thanks for spotting and pointing out the typo.

In the information leaking experiment, each multivariate one-step observation $x_t$ is split into two vectors $x_t^a$ and $x_t^b$.
Computational, we first summarize the historical information before time step $t$ with an RNN and denote it as $h_t =$
$f(\mathbf{x}_{<t}) := \mathrm{RNN}(x_{t-1}, h_{t-1})$. Then, $p(x_t \mid \mathbf{x}_{<t}) = p(x_t^a \mid \mathbf{x}_{<t})p(x_t^b \mid x_t^a, \mathbf{x}_{<t})$ are parameterized as two multivariate
Gaussians: $p(x_t^a \mid \mathbf{x}_{<t}) := \mathcal{N}(x_t^a; \mu_a(h_t), I\sigma_a^2(h_t))$ and $p(x_t^b \mid x_t^a, \mathbf{x}_{<t}) := \mathcal{N}(x_t^b; \mu_b(h_t, x_t^a), I\sigma_b^2(h_t, x_t^a))$, where
$\mu_a, \sigma_a$ and $\mu_b, \sigma_b$ are all trainable MLPs that output the vector-valued mean and (diagonal) variance for the corresponding
distributions. Hence, $x_t^a$ is treated as a vector of dimension $|x_t^a|$ instead of a sequence when fed into $\mu_b, \sigma_b$.

In our experiments, by decreasing $L$, the gap between F-SRNN and F-RNN decreases, and gradually F-RNN outperforms
F-SRNN. However, by using the RNN-hier architecture in our paper, the deterministic RNN model outperforms the
SRNN model in the settings with any $L$ value.

We will add citations in Table 4. We haven't conducted experiments in language modeling and image density estimation
tasks. But from existing publications, the state-of-the-art results of these tasks are produced by auto-regressive style
models.

**To Reviewer 3:**

Thanks for your suggestion on comparing the running time of different models. We will include this part of the results
in the revised version. The running times of training models for 40k updating steps on TIMIT are summarized in Table
1.

| Input Length | | 8000 | | | | | | 1000 |
|---|---|---|---|---|---|---|---|---|
| Model Name | F-RNN | F-SRNN | $\delta$-RNN | RNN-hier | SRNN-hier | RNN-flat | SRNN-flat | RNN-hier |
| Training Time | 0.54h | 0.94h | 0.90h | 9.92h | 12.52h | 37.48h | 42.26h | 1.7h |
| Log-Likelihood | 32,745 | 69,296 | 66,453 | 109,641 | 107,912 | 117,721 | 109,284 | 101,713 |

Table 1: Training time comparison between various models.

Admittedly, modeling the intra-step correlation would require extra computation time. Hence, this leads to a trade-off
between quality and speed. Ideally, latent-variable models would provide a solution close to the sweet point of this
trade-off. However, in our experiment, we find a simple hierarchical auto-regressive model trained with a shorter
input length could already achieve significantly better performance with a comparable computation time (RNN-hier vs.
F-SRNN in Table 1). We will add this discussion in the revised version.

Finally, we would like to emphasize that the goal of this work is to perform a fair and informative reexamination
of recurrent stochastic models rather than downplay any model. Based on our analysis and empirical evidence, we
hope to (1) correct the previous misleading conclusion that SRNN can already achieve better results compared with
deterministic RNN in the sequential density estimation, (2) provide a more realistic benchmark with SOTA baselines
for speech density estimation and encourage future researchers to perform a more meaningful model comparison, (3)
offer some informative analysis and understanding of what SRNN is actually doing in practice. Overall, we may still
have a long way to go to really fulfill the theoretical advantage of stochastic sequential models.

**To Reviewer 4:** We think the massive improvement provided by the auto-regressive model (including column 2 and
other columns) shows that the performance of the deterministic model is heavily underestimated in the previous biased
experiment setting.

We are not entirely sure about the motivation of the multi-frame setting. One possibility is to simulate the case of
modeling natural multi-variate sequences such the midi music. The computation speed could be another consideration
because the sequence length of speech data is much longer than language and image data, whose sample rate is 16k per
second.

We have not conducted in-depth research on different sample rates yet. According to popular speech synthesis papers,
WaveNet uses 16k sample rate and DeepVoice uses 16k and 48k.

[Meta-Review · NeurIPS 2019]

This paper was reviewed by three expert reviewers and received two Accept and one Weak Accept recommendations. After rebuttal, all the three reviewers are positive about this paper, and agree that this paper is well written, well motivated and clearly presented. The experiments are also carefully conducted, and the results are convincing. The biggest concern at the beginning was raised by R3, who questioned about the context of the results, runtimes and the trade-off between computational cost and performance. The rebuttal successfully addresses R3's concerns. Therefore, the AC recommends accepting the paper.